# CryptoNAS: Private Inference on a ReLU Budget

**Zahra Ghodsi, Akshaj Veldanda, Brandon Reagen, Siddharth Garg**
New York University
{zg451, akv275, bjr5, sg175}@nyu.edu

## Abstract

Machine learning as a service has given raise to privacy concerns surrounding clients' data and providers' models and has catalyzed research in private inference (PI): methods to process inferences without disclosing inputs. Recently, researchers have adapted cryptographic techniques to show PI is possible, however all solutions increase inference latency beyond practical limits. This paper makes the observation that existing models are ill-suited for PI and proposes a novel NAS method, named CryptoNAS, for finding and tailoring models to the needs of PI. The key insight is that in PI operator latency costs are inverted: non-linear operations (*e.g.*, ReLU) dominate latency, while linear layers become effectively free. We develop the idea of a *ReLU budget* as a proxy for inference latency and use CryptoNAS to build models that maximize accuracy within a given budget. CryptoNAS improves accuracy by $3.4\%$ and latency by $2.4\times$ over the state-of-the-art.

## 1   Introduction

User privacy has recently emerged as a first-order constraint, sparking interest in *private inference* (PI) using deep learning models: techniques that preserve data *and* model confidentiality without compromising user experience (*i.e.*, inference accuracy). In response to this concern, highly-secure solutions for PI have been proposed using cryptographic primitives, namely with homomorphic encryption (HE) and secure multi-party computation (MPC). However, both solutions incur substantial slowdown to the point where even state-of-the-art techniques that combine HE and MPC cannot achieve practical inference latency. Most prior work has focused on developing new systems (e.g., Cheetah [1]) and security protocols (e.g., MiniONN[2]) for privacy, while little effort has been made to find new network architectures tailored to the needs of PI. Realizing practical PI requires a better understanding of latency bottlenecks and new deep learning model optimizations to address them directly.

Prior work on PI [3, 2, 4] has developed cryptographic protocols optimized for common CNN operators. High-performance protocols take a hybrid approach to combine the strengths–and avoid the pitfalls–of different cryptographic methods. Most protocols today use one method for linear (e.g., secret sharing [5] for convolutions) and another for non-linear layers (e.g., Yao's garbled circuit [6] for ReLUs). Figure 1 compares the cryptographic latency of ReLU and linear layers for the MiniONN PI protocol [2] (details in Section 2). The data highlights how a layer's "plaintext" latency has little bearing on its corresponding cryptographic latency: ReLU operations dominate latency, taking up to $10,000\times$ longer to process than convolution layers. Given the inversion of operator latency costs, enabling efficient PI requires developing new models that maximize accuracy while minimizing ReLU operations, which we refer to as a model's *ReLU budget*. This stands in stark contrast to existing neural architecture search (NAS) approaches that focus only on optimizing the number of floating point operations (FLOPs).

Based on this insight we propose two optimization techniques: *ReLU reduction* and *ReLU balancing*. ReLU reduction describes methods to reduce the ReLU counts starting from a ReLU-heavy baseline network, for which two solutions are developed: ReLU pruning and ReLU shuffling.

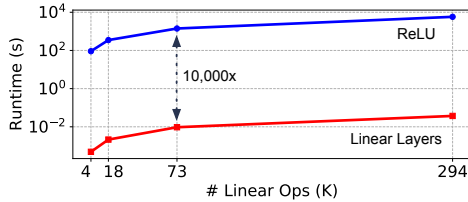

Figure 1: PI latency of a convolution and following ReLU layer. The convolution layer has input size $8 \times W \times W$ with $W \in \{8, 16, 32, 64\}$ and filter size $8 \times 3 \times 3$.

ReLU pruning removes unnecessary ReLUs from existing models, specifically from skip-connection and re-shaping layers of CNNs. ReLU shuffling moves the skip connection base such that the forwarded activations are post-ReLU, eliminating the redundancy. We show ReLU reduction is effective and can be implemented with negligible accuracy impact, e.g., ReLU pruning reduces a model's ReLU count by up to $5.8\times$ and reduces latency proportionately. ReLU balancing scales the number of *channels* in the CNN to maximize model capacity constrained by a ReLU budget. We prove formally that the maximum model capacity under a ReLU budget is achieved when *constant* ReLUs per layer are used; in contrast, traditional channel scaling commonly assumes constant FLOPs per layer [7, 8, 9]. ReLU balancing is most effective for smaller models, which benefit the most from the additional parameters, providing an accuracy improvement of up to $3.75\%$ on the CIFAR-100 dataset.

Leveraging these optimizations, we propose CryptoNAS: a new and *efficient* automated search methodology to find models that maximize accuracy within a given ReLU budget. CryptoNAS searches over a space of feed-forward CNN architectures with skip connections (referred to as the *macro-search* space in NAS literature [9, 10]). First, to incorporate constrained ReLU budgets, CryptoNAS uses ReLU reduction to make skip connections effectively "free" in terms of latency. Next, the number of channels in the *core network*, defined as the skip-free layers, are scaled based on ReLU balancing, which keeps the number of ReLUs per layer constant. Finally, CryptoNAS uses an efficient *decoupled* search procedure based on the observation that PI latency is *dominated* by the ReLUs of the core network, while skips can be freely added to increase accuracy. CryptoNAS uses the ENAS [9] macro search to determine where to add skips, and *separately* scales the size of the core network to meet the ReLU budget.

This paper makes the following contributions:

- Develop the idea of a ReLU budget, showing how ReLUs are the primary latency bottleneck in PI to motivate the need for new research in ReLU-lean networks.
- Propose optimizations for ReLU-efficient networks. ReLU reduction (ReLU pruning and shuffling) reduce network ReLU counts and ReLU balancing (provably) maximizes model capacity per ReLU. Our ReLU reduction technique reduces ReLU cost by over $5\times$ with minimal accuracy impact, while ReLU balancing increases accuracy by $3.75\%$ on a ReLU budget.
- We develop CryptoNAS to automatically and efficiently find new models tailored for PI. CryptoNAS decouples core and PI optimizations for search efficiency. Resulting networks along the accuracy-latency Pareto frontier outperform prior work with accuracy (latency) savings of $1.21\%$ ($2.2\times$) and $4.01\%$ ($1.3\times$) on CIFAR-10 and CIFAR-100 respectively.

## 2 Background on Private Inference

CryptoNAS implements the MiniONN protocol [2] for PI, using the same threat model and security assumptions. We note, however, that the choice of PI protocol is largely independent of the qualitative results; e.g., we expect similar benefit if secret sharing is replaced with HE, as done in Gazelle [3]. In this section we provide the necessary background for MiniONN and cryptographic primitives used.

### 2.1 Cryptographic Primitives

MiniONN uses the forllowing two-party compute (2PC) cryptographic primitives: secret sharing for linear layers and garbled circuits for non-linear operators. The protocol is executed between two parties: a server ($S$), hosting the model, and a client ($C$) with the input data for inference using $S$'s model.

**Secret Sharing** is a 2PC technique for computing additions and multiplications on secure data; MiniONN uses the SPDZ version of secrete sharing [11]. Imagine that $C$ and $S$ wish to compute

the dot-product operation $y = \mathbf{w} \cdot \mathbf{x}$, where $\mathbf{x} \in \mathbb{F}_p^{n \times 1}$, an $n$-dimensional column vector in finite field, $\mathbb{F}_p$, (defined by prime $p$) belongs to $C$. Similarly, row vector $\mathbf{w} \in \mathbb{F}_p^{1 \times n}$ belongs to $S$. To do so, $C$ first "blinds" $\mathbf{x}$ using a randomly picked vector $\mathbf{r} \in \mathbb{F}_p^{n \times 1}$ and sends $\mathbf{x}^S = \mathbf{x} - \mathbf{r}$ to $S$. $\mathbf{x}^S$ is referred to as $S$'s "share." $C$'s share is simply $\mathbf{x}^C = \mathbf{r}$. Note that the sum of the client and server shares equals the secret, i.e., $\mathbf{x}^C + \mathbf{x}^S = \mathbf{x}$ — this will hold true for any value shared between the two parties. Note also that $S$ cannot infer *anything* about $\mathbf{x}$ from its share alone.

To handle multiplications, the SPDZ protocol uses Beaver's multiplication triples ($<u, v, \mathbf{w} \cdot \mathbf{r}>$), that are pre-computed off-line with $\mathbf{w}$ and $\mathbf{r}$ as inputs from $S$ and $C$ respectively. The protocol generates and outputs $u \in \mathbb{F}_p$ to $S$ and $v \in \mathbb{F}_p$ to $C$, such that (1) $v$ is a random value and no party learns the other party's output; and (2) $u + v = \mathbf{w} \cdot \mathbf{r}$. Using the triple, $S$ can compute, $y^S$, its share of the result $y$ as follows: $y^S = \mathbf{w} \cdot \mathbf{x^S} + u$. $C$'s share is simply $y^C = v$. One can verify that $y^S + y^C = y = \mathbf{w} \cdot \mathbf{x}$.

In practice, multiplication triples can be generated using HE. Although HE is computationally expensive, this step can be performed *offline* if $S$'s input ($\mathbf{w}$) is fixed — note that the triples only require $C$'s random input $\mathbf{r}$, which can also be generated offline. By moving triple generation offline, the protocol's *online* costs of performing a linear operation (e.g., a dot-product) is reduced to near plain-text performance.

**Yao's Garbled Circuits (GC)** Secret sharing only supports additions and multiplications, it cannot be used for non-polynomial non-linear operations like ReLU. GC is an alternate secure 2PC solution that, unlike secret sharing, works on *Boolean* representations of $C$'s and $S$'s inputs and allows the two parties to compute a bi-variate Boolean function $B : \{0, 1\}^n \times \{0, 1\}^n \to \{0, 1\}$ without either party revealing its input to the other. We briefly describe the GC protocol and refer the reader to [12] for more details.

The GC protocol represents function $B$ as a Boolean logic circuit with 2-input logic gates. $C$ and $S$ "evaluate" the circuit gate by gate; critically, each gate evaluation involves data transfer between the parties and expensive cryptographic computations, i.e., AES decryptions. Thus, as observed in Figure 1 the GC protocol incurs significant online cost per function evaluation. In practice, the GC protocol is asymmetric in that one party acts as the so-called "garbler" and the other as the "evaluator." The evaluator obtains the encrypted output and shares it with the garbler. The encrypted output is then decrypted by the garbler.

## 2.2 The MiniONN Protocol for Private Inference

Here we briefly summarize the MiniONN protocol and refer the reader to the original paper for details [2]. MiniONN employs secret sharing and GC for private inference in a client-server ($C$-$S$) setting where $C$ and $S$ keep their inputs and model private, respectively. Building on MiniONN's terminology, we model a neural network containing $D$ layers as:

$$\mathbf{y_{i+1}} = f\left(\mathbf{W_i} \cdot \mathbf{y_i} + \mathbf{b_i}\right) \quad \forall i \in [0, D-1], \tag{1}$$

where $\mathbf{y_0} = \mathbf{x}$ is $C$'s input and $y_D = z$ is the output prediction. $\{\mathbf{W_i}\}_{i \in [0, D-1]}$ and $\{\mathbf{b_i}\}_{i \in [0, D-1]}$ are $S$'s model weights and biases, respectively, and $f$ is the ReLU function.

Imagine that $S$ and $C$ have shares of layer $i$'s inputs, $\mathbf{y_i^S}$ and $\mathbf{y_i^C}$, respectively. MiniONN first uses secret sharing to compute shares of $\mathbf{q_i} = \mathbf{W_i} \cdot \mathbf{y_i} + \mathbf{b_i}$, which we will call $\mathbf{q_i^S}$ and $\mathbf{q_i^C}$. Recall that because multiplication triples are generated offline, this computation happens almost as quickly as it would non-securely. Finally, the GC protocol is used to privately perform the ReLU operation with $S$ as the garbler and $C$ as the evaluator. $C$ contributes $\mathbf{q_i^C}$ and $\mathbf{r_{i+1}}$, its pre-generated share of the subsequent layer's input, while $S$ contributes $\mathbf{q_i^S}$. The GC protocol computes $\mathbf{y_{i+1}^S} = max(0, \mathbf{q_i^C} + \mathbf{q_i^S}) - \mathbf{r_{i+1}}$; only $S$ is able to decrypt this value and learn the output. $C$ sets $\mathbf{y_{i+1}^C} = \mathbf{r_{i+1}}$. It can be verified that $C$ and $S$ now hold shares of layer $i$'s outputs. The protocol proceeds similarly for the next layer.

## 3 CryptoNAS: Finding Accurate, Low-Latency Networks for PI

We now present CryptoNAS: a method for finding accurate CNNs on a ReLU budget. A formal NAS procedure described in Section 3.3, which comprises network optimizations to reduce ReLU counts

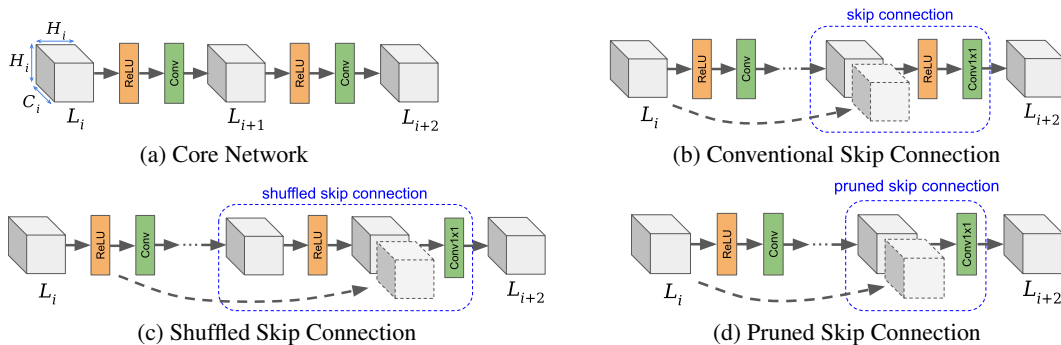

(a) Core Network

(b) Conventional Skip Connection

(c) Shuffled Skip Connection

(d) Pruned Skip Connection

Figure 2: Illustration of ReLU count reduction methods. (a) Core network prior to adding skip connections. (b) Conventional skip connections increase ReLU counts. (c) ReLU shuffling preserves the functionality of (b) but with lower ReLU cost. (d) ReLU pruning further reduces ReLU cost by completely eliminating ReLUs from skips.

(Section 3.1) and maximize network parameters per ReLU (Section 3.2). Updates to the MiniONN protocol to support skip connections are described in Section 3.4.

**CryptoNAS Search Space** As in prior work [10, 9], CryptoNAS searches over all feed-forward depth $D$ CNN architectures where layer $i \in [0, D-1]$ has an input feature map of shape $H_i \times H_i \times C_i$ and performs standard convolutions with $C_{i+1}$ filters of size $F_i \times F_i \times C_i$. Layer $i$ can take inputs from any prior layer; the channels of the corresponding feature maps are concatenated and reshaped to the appropriate input dimension using $1 \times 1$ convolutions, as shown in Figure 2. Vector $\mathbf{S_i} \in \{0, 1\}^{i-1}$ indicates the preceding layers from which layer $i$ takes inputs. Consequently, the design space for CryptoNAS consists of the depth $D$ and per-layer parameters $\{H_i, C_i, F_i, \mathbf{S_i}\}_{i \in [0, D-1]}$.

## 3.1 Reducing ReLU Counts

We find that many existing hand-crafted deep learning models and NAS techniques use ReLUs indiscriminately — this is not surprising, as in plaintext inference ReLU latency is far less than MAC operations and thus effectively free. We propose two simple optimizations for skip-based CNN layers to (i) significantly reduce ReLUs while preserving accuracy (ReLU Shuffling) or (ii) saving even more ReLUs in return for a small drop in accuracy (ReLU Pruning).

**ReLU Shuffling** A standard way (e.g., as done in DenseNets [13] and ENAS [9]) to add skip connections is to forward the post-convolution output of a layer and concatenate it with the convolution output of a future layer — the concatenated feature maps are then passed through a ReLU operation and re-shaped, as shown in Figure 2. Thus, the number of ReLU operations in a layer is magnified by the number of its incoming skip connections. ReLU shuffling proposes to move (shuffle) the standard convolution-skip-ReLU sequence to convolution-ReLU-skip (see Figure 2(c)). In doing so, the ReLU after the first convolution in the core network is *reused* by the skip-connection. In PI ReLU shuffling is strictly beneficial: it precisely preserves network functionality to achieve the exact same accuracy with fewer ReLUs.

**ReLU Pruning** With ReLU shuffling the cost of skip connections are reduced but skips still add ReLUs to the core network. ReLU pruning goes a step further by *removing* all ReLUs introduced by skips, as shown in Figure 2(d). We found that these ReLUs can be removed with only minor accuracy impact. ReLU pruning further reduces ReLU counts by $2\times$ over ReLU shuffling and we show in Section 4 that for smaller ReLU budgets, it actually provides higher accuracy compared to ReLU Shuffling while being competitive for higher ReLU budgets.

## 3.2 ReLU Balancing for Maximum Network Capacity

Many traditional networks (e.g., VGG [8], ResNet [7]) and NAS procedures [9, 10] keep the number of FLOPs constant across layers — i.e., when the spatial resolution is reduced across pooling layers (typically by a factor of two in each dimension), the number of channels is correspondingly increased

(typically doubled)[1]. CryptoNAS's optimization objective, i.e., to minimize ReLUs and not FLOPs, suggests an alternate channel scaling rule wherein the number of ReLUs is kept constant across layers. Indeed, we show that this rule is not only intuitively appealing, but also maximizes model capacity (number of trainable parameters) for a fixed ReLU budget[2].

**Lemma 3.1.** *Consider a depth $D$ CNN with $H_i \times H_i \times C_i$ input tensors and $F \times F \times C_i$ sized convolutional filters in layer $i$ for all $i \in [0, D-1]$ followed by a ReLU operation. Assuming the spatial resolution is reduced by a factor of $\alpha$ in each pooling layer, the number of trainable parameters given a fixed ReLU budget is maximized if $C_{i+1} = \alpha^2 C_i$ after pooling layer $i$.*

We show in Section 4 that ReLU balancing provides consistent wins in accuracy over traditional FLOP balanced networks, especially for small ReLU budgets.

---

**Algorithm 1** CryptoNAS Algorithm

---

1: **Input:** $\mathbb{D} \coloneqq$ Set of network depths, $R_{\text{bdg}} \coloneqq$ ReLU Budget
2: **Output:** $\mathcal{N}^* \coloneqq$ Highest accuracy model under $R_{\text{bdg}}$
3: **Definitions:**
4: $C_i, H_i, F_i, \boldsymbol{S}_i \coloneqq$ Channels, resolution, filter size and skip connections of layer $i$
5: $\alpha \coloneqq$ resolution scaling factor
6: **function** CRYPTONAS($\mathbb{D}, R_{\text{bdg}}$)
7:     **for** $D \in \mathbb{D}$ **do**
8:         /* Optimize core network parameters
9:         
$$H_i^* = \begin{cases} H_0, & \text{if } 0 < i \le D/3 \\ H_0/\alpha, & \text{if } D/3 < i \le 2D/3 \\ H_0/\alpha^2, & \text{otherwise} \end{cases} \quad C_i^* = \begin{cases} R_{\text{bdg}}/DH_0^2, & \text{if } 0 < i \le D/3 \\ \beta C_1, & \text{if } D/3 < i \le 2D/3 \\ \beta^2 C_1, & \text{otherwise} \end{cases}$$
10:
11:         $\{\boldsymbol{S}_i^*, F_i^*\}_{i \le L} \leftarrow$ ENAS($D$) /* Run ENAS to optimize skip connections and filter sizes
12:
13:         $\mathcal{N}_D^* = \{\{C_i^*, H_i^*, F_i^*, \boldsymbol{S}_i^*\}\}_{i=0,\dots,D-1}$
14:     $\mathcal{N}^* = \arg\max_{D \in \mathbb{D}} acc(\mathcal{N}_D^*)$
15:     **return** $\mathcal{N}^*$

---

## 3.3 The CryptoNAS Method

CryptoNAS uses the above optimizations to develop a NAS technique tailored to the needs of PI, searching over a *large* space of CNNs with skip connections. Any architecture in this search space can be viewed as containing two components: the core network (see Figure 2(a)) and the skip connections between layers of the core. For a traditional FLOP constrained search, such as in [14], these components are *coupled*: the FLOPs in a larger core network would have to be compensated with fewer skips and vice-versa, requiring an expensive search over the product of these two sub-spaces.

Instead, CryptoNAS leverages the two optimizations described in Sections 3.1 and 3.2 to efficiently decouple the search, i.e., to search for the core network and skip connections *separately*. That is, since skips are effectively free and the PI latency is dominated by the core network, searching for the most accurate architecture under a ReLU budget breaks down into two separate questions: (i) what is the most accurate core network within a ReLU budget and (ii) where should skips be added to maximally increase the core network's accuracy?

CryptoNAS answers these questions for networks of varying depth $D$ picked from a set $\mathbb{D}$ specified by the user (line 7), as described in Algorithm 1. For a given depth, CryptoNAS first determines the core network's parameters as follows. Consistent with a body of prior work on NAS, pooling layers are inserted at layers $\frac{D}{3}$ and $\frac{2D}{3}$ to reduce spatial resolution by a factor of $\alpha = 2$ in each dimension. Following the ReLU balancing rule (Section 3.2), the number of channels is correspondingly increased by a factor of $\alpha^2 = 4$ after each of the two pooling layers. With this in place, the only remaining free variable is $C_1$, the number of channels in the first layer; $C_1$ (and correspondingly the number of channels in all subsequent layers) is set to the largest values such that total ReLU count of the core network is within the budget (line 9).

Separately, CryptoNAS uses ENAS macro-search [9] to determine the best skip connections for each depth (line 11). ENAS is ideally suited for this purpose because the search procedure depends on depth only and not the number of channels in the core model (recall that we seek to separately scale the size of the core model). The skip connections returned by ENAS are then stitched into the previously found core model. These two steps yield optimized networks for different depths. The final step selects the model with the highest accuracy on the validation set (line 14). Experiments show that CryptoNAS produces a Pareto frontier of points that outperform existing solutions (see Section 4.2). Critically, for each point on the Pareto front, CryptoNAS performs only one ENAS search followed by a single run to train the core model plus skips for each network depth explored.

### 3.4 Incorporating Skip Connections in MiniONN

The MiniONN protocol only considers skip-free CNNs. Here we extend MiniONN to support skip connections, which are necessary for finding highly-accurate models. First, assume there is a skip connection between layer $i$ from the previous layer. Recall that skip connections are made by concatenating the outputs of ReLU layers and reshaping the concatenation via a $1\times1$ convolution. Thus, we can write the output of this concatenation layer as

$$\mathbf{y_{i+1}} = \mathbf{W'_{i+1}} \cdot [\mathbf{y'_{i+1}}; \mathbf{y_i}] + \mathbf{b'_{i+1}} \tag{2}$$

where $\mathbf{W'_{i+1}}$ and $\mathbf{b'_{i+1}}$ are the weights and biases of the $1\times1$ convolution layer, and $\mathbf{y'_{i+1}} = f(\mathbf{W_i} \cdot \mathbf{y_i} + \mathbf{b_i})$. To compute this layer privately, $S$ and $C$ can generate multiplication triples offline for weight $\mathbf{W'_{i+1}}$ and random value $[\mathbf{y'^C_{i+1}}; \mathbf{y^C_i}]$ (recall that the protocol takes these two inputs). The random value in the multiplication triple protocol comes from $C$ and is constructed by concatenating $C$'s share of the two layers, which are determined in the offline phase as well. This construction is easily extended to skip connections for multiple layers.

## 4 Evaluation

In this section we evaluate CryptoNAS and show that it outperforms the SOTA, analyze scaling trends, and present a case study to show our optimizations generalize to architectures, beyond the core CryptoNAS search space, e.g., WRNs [15].

### 4.1 Experimental Setup

CryptoNAS implements MiniONN [2] for PI with extensions described in Section 3.4. Beaver triples are generated with the SEAL [16] library and the ABY library [17] is used for GC. We report online runtimes for the client and server together, including both computation and communication costs. Since ReLU evaluations are bottlenecks for both computation and communication, reductions in ReLU counts reduce both costs proportionally. CryptoNAS is evaluated on CIFAR-10 and CIFAR-100 [18]. We note that CIFAR-10/100 is the standard for both PI and NAS research due to the long run times, and we chose them to be consistent with prior work. Datasets are preprocessed with image centering (subtracting mean and dividing standard deviation), and images are augmented for training using random horizontal flips, 4 pixel padding, and taking random crops. Experiments for latency are run on a 3 GHz Intel Xeon E5-2690 processor with 60GB of RAM, and networks are trained on Tesla P100 GPUs. We use CryptoNAS to discover three models with depth=$\{6, 12, 24\}$, which we refer to as CNet1, CNet2 and CNet3.

### 4.2 Results

**ReLU shuffling and pruning reduce ReLUs with minimal impact on accuracy.** Table 1 compares the proposed ReLU shuffling (Figure 2(c)) and ReLU pruning (Figure 2(d)) optimizations for different ReLU budgets on the CNet2 model. The ENAS baseline (Figure 2(b)) version of these models have $385K$, $1.92M$, and $3.89M$ ReLUs; the accuracy of the ENAS baseline is the same as obtained via ReLU shuffling. We observe that compared to baseline, ReLU shuffling reduce ReLU counts by about $1.9\times$ for this model (across all modes we see up to $4.76\times$ drop in ReLU counts) without compromising accuracy. For the same reduction in ReLU counts, ReLU pruning provides **2.2% *higher* accuracy** for CIFAR-100 at a 100K ReLU budget, but incurs a small accuracy drop for larger ReLU budgets (0.64% for 1M budget).

Table 1: Comparing the accuracy of networks generated by CryptoNAS for different ReLU budgets achieved by ReLU pruning and shuffling.

| Model (ReLU-Budget) | CIFAR-10 | | | | CIFAR-100 | | | |
|---|---|---|---|---|---|---|---|---|
| | ReLU Pruning | | ReLU Shuffling | | ReLU Pruning | | ReLU Shuffling | |
| | Params | Acc | Params | Acc | Params | Acc | Params | Acc |
| CNet2-100K | 1.2M | 92.18% | 305K | 90.71% | 1.3M | 68.67% | 320K | 66.46% |
| CNet2-500K | 30M | 94.41% | 7.6M | 94.51% | 31M | 77.2% | 7.8M | 77.69% |
| CNet2-1M | 124M | 95.00% | 30M | 95.47% | 128M | 79.07% | 31M | 79.71% |

Table 2: Results comparing channel scaling techniques across datasets (CIFAR10/100) with 3 models depths (6, 12, 24). ReLU balancing produces the most accurate models on a ReLU budget.

| Model (Base-Dataset) | ReLU Budget | Depth | ReLU Balanced | | FLOP Balanced | | Channel Balanced | |
|---|---|---|---|---|---|---|---|---|
| | | | Params | Acc | Params | Acc | Params | Acc |
| CNet1-C10 | 86K | 6 | 1.4M | 91.28% | 352K | 90.55% | 94K | 88.07% |
| CNet2-C10 | 344K | 12 | 15M | 94.04% | 3.5M | 93.98% | 1M | 93.07% |
| CNet3-C10 | 1376K | 24 | 167M | 95.55% | 39M | 95.49% | 9M | 95.09% |
| CNet1-C100 | 86K | 6 | 1.3M | 68.13% | 313K | 64.38% | 94K | 50.14% |
| CNet2-C100 | 344K | 12 | 15M | 75.64% | 3.8M | 74.05% | 1.2M | 69.21% |
| CNet3-C100 | 1376K | 24 | 142M | 79.59% | 34M | 79.23% | 9M | 77.00% |

**ReLU balanced scaling maximizes accuracy.** Next we evaluate our channel scaling rule optimization, ReLU balancing, against the commonly used FLOP balancing approach and a strawman "channel balancing" solution that has the same number of channels in each layer. Our results are shown in Table 2. We find that ReLU balancing *always* outperforms competing scaling techniques, likely because of its significantly larger model capacity under ReLU constraints. For CNet1-C100 at an 86K ReLU budget, ReLU balancing **increases accuracy by 3.75%** compared to traditional FLOP balancing. The benefits are most significant at lower ReLU budgets likely because these models underfit and benefit more from increased capacity.

**CryptoNAS perform better than maximal networks.** Given that skips are free, another relevant baseline can be a "maximal network" that includes all skips and largest filter sizes. We compare against maximal networks in Table 3. We observe that the maximal network for CNet3 ReLU balanced had an accuracy loss of 0.24% (CIFAR-10) and 0.48% (CIFAR-100) compared to the models found by

Table 3: Comparing networks discovered by ENAS to maximal networks in search space (all-skip, 5×5 filters).

| Model (Base-Dataset) | ENAS Arch | | Largest Arch | |
|---|---|---|---|---|
| | Params | Acc | Params | Acc |
| CNet3-C10 | 166M | 95.55% | 311M | 95.31% |
| CNet3-C100 | 149M | 79.59% | 311M | 79.11% |

ENAS. That is, the models returned by ENAS are more accurate than the largest all-skip models. This also reaffirms the results reported in [10], where models with all possible skip connections showed an accuracy drop.

**Efficiency of CryptoNAS search procedure.** The total runtime of CryptoNAS for a given ReLU budget involves a single ENAS run to find filter sizes and skip connections and a single training of the scaled core model with skip connections added per network depth explored. Running ENAS on CNet3 (our largest model) for CIFAR-100 dataset takes about 22hrs on relatively slow P100 GPUs, and final training takes about 19hrs.

**CryptoNAS models outperform prior art across the accuracy-latency Pareto frontier.** Using CryptoNAS, we sweep a range of model depths and ReLU budgets to understand latency-accuracy tradeoffs in PI and plot the Pareto points in Figure 3 (CryptoNAS in red). We compare CryptoNAS against prior work that (a) achieve state-of-art performance using standard augmentation and ReLU activations including ResNets [7] and DenseNets [13], (b) networks like MobileNets-v2 [19] that are optimized for FLOPs instead of ReLUs, (c) the ENAS-macro and ENAS-micro search results [9], and (d) networks from prior work on PI including MiniONN [2] and Delphi [20].

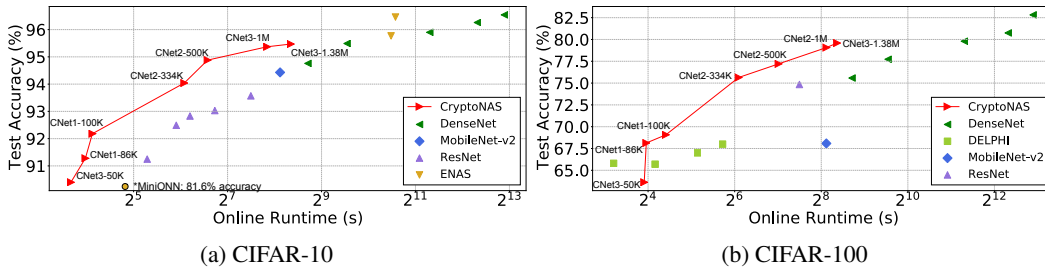

(a) CIFAR-10                                        (b) CIFAR-100

Figure 3: PI runtimes for CryptoNAS networks compared to state-of-the-art on CIFAR-10 and CIFAR-100 datasets. The runtime is the sum of client and server online costs.

We see that CryptoNAS's Pareto frontier dominates all prior points for regions of interest. On CIFAR-10, CNet2-100K has $0.93\%$ higher accuracy while being $2.2\times$ faster on inference latency compared to ResNet-20, and a more than $10\%$ accuracy improvement over MiniONN. Compared to ResNet-110, CNet3-500K is $1.9\times$ faster with $1.31\%$ higher accuracy. CryptoNAS compares even more favorably to MobileNet-v2; CryptoNAS is $1.2\times$ faster with $1.31\%$ higher accuracy. On CIFAR-100, CNet2-1M is $3.49\%$ more accurate than DenseNet-12-40 while being $1.5\times$ faster. We also achieve gains over Delphi, with CNet-100K being $1.6\times$ and $2.4\times$ faster than Delphi with 200K and 300K ReLU budget respectively and $2.07\%$ and $1.07\%$ more accurate.

We note that while Delphi does better than CryptoNAS for the smallest (50K) ReLU budget, it does so by systematically replacing a fraction of ReLUs with quadratic activations. In our evaluations, we give the benefit of considering all non-ReLU activations free. Second, we note that Delphi's optimizations can be applied over models found by CryptoNAS, replacing ReLUs in our networks with quadratics might yield further benefits. Finally, we note that the largest models from prior work provide slightly higher accuracy, but at *impractically* high PI latency of more than 30 minutes per inference. Down scaling these models to smaller sizes resulted in much lower accuracy than comparable CryptoNAS networks.

**CryptoNAS optimizations generalize to other network architectures.** We note that although we have thus far applied CryptoNAS's optimizations to ENAS-like CNN models, the optimizations are not ENAS specific and can be used to improve other models as well. As an example, we implemented ReLU balanced channel scaling to wide residual networks (WRN) [15] and found consistent improvements in accuracy relative to the baseline FLOP balanced WRNs for iso-ReLU budgets. Despite the more aggressive baseline, CryptoNAS still outperforms re-scaled WRN across a range of ReLU budgets, see Table 4 in the Appendix for complete results.

## 5  Related Work

In this section we discuss prior work on PI and NAS as related to CryptoNAS. CryptoNets [21], one of the earliest works on PI used fully homomorphic encryption (FHE) to guarantee data (but not model) privacy. Due to its reliance on FHE for all network layers, CryptoNets is restricted to using only polynomial activations. Subsequent work, including MiniONN [2], SecureML [4], Gazelle [3] and Delphi [20] seek to provide both data and model privacy and support standard activations like ReLUs; to do so, they switch between separately optimized protocols for linear layers and non-linear layers. Common to these protocols is the use of expensive GC for ReLU activations. As such, we expect CryptoNAS's benefits to extend to these protocols as well. In addition to cryptographic optimizations, Delphi [20] also proposes to automatically replace a subset of ReLUs with quadratic activations using an automated search procedure. While CryptoNAS outperforms Delphi for all but one data-point, Delphi's optimizations can be applied over models found by CryptoNAS for further benefit and would be an interesting direction for future research. Nonetheless, we note that large quadratic networks are hard to train [22]. A separate line of work, DeepSecure [23] and XONN [24], has proposed PI methods for binarized neural networks but these typically have lower accuracy than conventional networks.

There is a large and growing body of work on NAS including methods that make use of Bayesian optimization [14], reinforcement learning (RL) [10], hill-climbing [25], genetic algorithms [26], etc. We refer the reader to an excellent survey paper by Elsken et al. [27] for more details. Most relevant

to this work are multi-objective NAS methods, for example, MnasNet [28], that uses measured model latency to optimize for inference time and accuracy. While these approaches could also be used to optimize for ReLU costs, they would still end up searching over spaces with a relatively high number of ReLUs per FLOP. The ReLU optimizations and decoupled search insights in CryptoNAS should help these approaches as well in terms of both accuracy and search efficiency.

# 6 Conclusions

This paper presents CryptoNAS, a new NAS method to minimize PI latency based on the observation that ReLUs are the primary latency bottleneck in PI. This observation motivates the need for new research in "ReLU-lean" networks. To this end, we have proposed optimizations to obtain ReLU-lean networks including ReLU reduction via ReLU pruning and shuffling, and ReLU balancing that (provably) maximizes model capacity per ReLU. ReLU reduction yields $> 5\times$ reductions in ReLU cost with minimal impact on accuracy, while ReLU balancing increases accuracy by $3.75\%$ on a fixed ReLU budget compared to traditional approaches. Finally, we develop CryptoNAS to automatically and efficiently find new models tailored for PI based on a decoupling of the search space. Resulting networks along the accuracy-latency Pareto frontier outperform prior work with accuracy (latency) savings of $1.21\%$ ($2.2\times$) and $4.01\%$ ($1.3\times$) on CIFAR-10 and CIFAR-100 respectively.

## Broader Impact

Privacy is an increasingly and critically important societal concern, especially given the fraying bonds of trust between individuals, organizations and governments. By enabling users to protect the privacy of their data and organizations to protect the privacy of their models without compromising accuracy and at reasonable latency, CryptoNAS seeks to herald a new suite of private inference solutions based on cryptography. On the other hand, performing computations in the encrypted domain can inadvertently affect the ability to detect other types of attacks on machine learning systems, e.g. inference attacks.

## Acknowledgments and Disclosure of Funding

This project was funded in part by NSF grants #1801495 and #1646671.

## Footnotes

[1]Recall that FLOPs are proportional to the square of number of channels.

[2]See Appendix for the complete proof.

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
