[Supplementary Material]

# A   Proof of ReLU Balancing Rule

We present the proof of Lemma 3.1 here.

*Proof.* We seek to optimize the number of trainable parameters, i.e.:

$$\max \sum_{i \in [0, D-1]} F^2 C_i^2$$

subject to a constraint on the total number of ReLUs, which we can write as:

$$\sum_{i \in [0, D-1]} \frac{W_0^2}{\alpha^{2i}} C_i \leq R_{bdg}.$$

To solve this problem, we maximize its Lagrangian:

$$\max \sum_{i \in [0, D-1]} F^2 C_i^2 - \lambda \left( \sum_{i \in [0, D-1]} \frac{W_0^2}{\alpha^{2i}} C_i - R_{bdg} \right)$$

Setting derivatives with respect to $C_i$ to zero, we get $C_i = \frac{\lambda W_0^2}{2F^2 \alpha^{2i}}$ and hence, $C_{i+1} = \alpha^2 C_i$.  □

# B   CryptoNAS Generalized to Other Networks

Table 4 explores ReLU balanced channel scaling for original wide residual networks, as well as models scaled to lower ReLU budgets (WRN-16-4 and WRN-16-2). We observe that ReLU balanced models have more parameters and improve accuracy compared to FLOP balanced models for the same ReLU budget.

Table 4: Exploring ReLU ballancing in Wide ResNet architectures.

| Model | ReLU Budget | FLOP Balanced | | ReLU Balanced | |
|---|---|---|---|---|---|
| | | Params | Acc | Params | Acc |
| WRN-16-2 | 245K | 703K | 94.06% | 2.6M | 94.89% |
| WRN-16-4 | 475K | 2.8M | 95.39% | 10.4M | 95.6% |
| WRN-16-8 | 933K | 11.0M | 95.73% | 42M | 96.01% |
| WRN-40-4 | 1392K | 8.9M | 95.47% | 36M | 96.28% |
| WRN-28-10 | 2310K | 36.5M | 96.00% | 144M | 96.28% |