[Reviews · NeurIPS 2020]

Review 1

Summary and Contributions: The paper describes a new ReLU count optimized neural architecture search. This matters, because most cryptography-based private inference techniques are limited in performance due to the challenge of evaluating ReLUs in a privacy preserving manner. The new NAS shows significant improvements in the number of ReLUs, while preserving high accuracy of the trained models. The authors described an extension to the MiniONN protocol that leverages the skip connections optimized with CryptoNAS.

Strengths: The paper is largely agnostic to the specific private inferencing technique used, and hence has broad applicability. Based on these results, it seems that optimizing for PI friendly NNs is a promising direction. The results are very convincing (Fig. 3) and well presented. The approach taken for CryptoNAS makes sense to me, but I'm not an expert in NAS so it is hard for me to evaluate its novelty as a contribution.

Weaknesses: What is the break-down of the run-time? Is it mostly communication or computation? I think a more thorough discussion of the challenges of ReLU might be in order. For example, different techniques for evaluating it have different challenges, and different performance trade-offs. Is the main problem communication cost, or computation cost?

Correctness: The claims seem to make sense from a private inferencing and cryptography point-of-view. However, I'm unable to properly evaluate the NAS component of this work.

Clarity: The paper is very clear. I had no problems reading it.

Relation to Prior Work: The prior work discussion is somewhat one-sided, focusing on the MiniONN results for the private inferencing task. This is understandable, because there might not be any comparable prior work on the NAS part.

Reproducibility: Yes

Additional Feedback: A couple typos: Line 147: "if far less" Line 107: these techniques CAN be used for non-linear operations, but not for *non-polynomial*. Line 196: "after at layers" Line 203: "because because" Table 1 caption: "acheived"


Review 2

Summary and Contributions: The paper explores the fact that relative cost of linear and non-linear computation is different between normal and privacy-preserving computation. Based on this, it presents an optimization for the latter that removes ReLU layers in deep neural network inference in order to reduce the cost while maintaining or even increasing accuracy.

Strengths: The observation underlying the work is very important in privacy-preserving machine learning, and the proposed solution seems appropriate. I'm not aware of any other work treating this issue as thorough is this one, and it certainly deserves the attention of the community.

Weaknesses: I cannot think of any major limitations. Post rebuttal and discussion: I'm happy with the minor corrections suggested by the authors, and I maintain my evaluation from a secure computation perspective.

Correctness: I don't have the background to judge the NAS technique, but the ReLU reduction techniques are straight-forward.

Clarity: The paper is well written with very few exceptions: l147: "isn't" l267: "qaudratic" Algorithm 1: \beta is not explained.

Relation to Prior Work: Yes.

Reproducibility: Yes

Additional Feedback: - Yao's 1986 paper is not a good reference for learning about garbled circuits because he only presented it in a talk. A more appropriate reference would be Bellare et al. at CCS '12. - A few references contain unnecessary curly brackets, e.g. "{USENIX}".


Review 3

Summary and Contributions: [Post-Rebuttal Updates] In their rebuttal, the authors' promised to to add more baselines and details about their search spaces to the paper, as well as to release the source code for their experiments. I believe that these adequately address my original concerns, and I've increased my overall score for the paper from 3 to 6. In particular: 1) The authors provided baselines where they compared the accuracies of models discovered using ENAS against the largest possible models in their search spaces (Table 1 of the rebuttal). Furthermore, they promised to add the control experiments to their paper. 2) They clarified the search space used for their experiments ("the same as ENAS-macro"), and promised to publicly release their architectures and code. [Initial Summary] The submission considers the problem of designing neural network architectures for image classification which have low inference times when performing cryptographic private inference using a previously proposed method called MiniONN. The authors argue that when using MiniONN, multiplication and addition are nearly free while ReLU operations are expensive; this is very different from inference on non-encrypted data, where multiply-adds tend to dominate the total runtime. They propose a combination of manual network modifications and Neural Architecture Search to find network architectures which have good tradeoffs between accuracy and number of ReLUs. The techniques are: 1) "ReLU Shuffling:" Manually changing the positions of certain ReLU layers so that ReLUs are applied to layers with fewer channels. 2) "ReLU Pruning:" Manually removing certain ReLU layers from the network entirely. 3) Manually adjusting the number of output channels in different layers of the network in order to maximize the ratio of total trainable parameters (which is used as an approximation of network capacity) to ReLU operations. 4) Running a Neural Architecture Search algorithm called ENAS to determine (a) the skip connection pattern, i.e., which of the first N layers of the network should be used as inputs to the (N+1)st layer, and (b) the kernel size of the convolution used in each layer of the network (e.g., 3x3 or 5x5). This procedure is repeated for many different architecture depths, generating a *set* of network architectures -- one for each depth. Experimental results are reported on CIFAR-10 and CIFAR-100. NOTE: I'll focus on the NAS-related and architecture-related aspects of this paper, since that's what I'm most familiar with.

Strengths: 1) Optimizing for MiniONN inference time seems like an interesting -- and probably novel -- problem, since the authors convincingly argue that certain properties which make a network run quickly on MiniONN (e.g., low number of ReLUs) are much more important than we'd typically see on a device such as a CPU or GPU. 2) The paper's proposed ReLU shuffling and pruning optimizations come with good intuitive motivations, and are well-explained. (Section 3.1 and Figure 2.) 3) The submission provides experiments (Section 4.2 and Table 2) to support their claim that by changing the number of output filters in different parts of the network, they can obtain a significantly better Accuracy/FLOPS tradeoff than by using the standard heuristic of doubling the number of filters whenever they halve the feature map height/width. 4) The submission measures the online runtime of their method and several others using MiniONN, and compares the inference time/accuracy tradeoff for their searched network architectures against a variety of existing methods: DenseNet, MobileNet-V2, ResNet, and models found using the ENAS architecture search algorithm. Experiments indicate that the authors' new network architectures significantly outperform competing methods.

Weaknesses: 1) Given my current understanding of the paper and the authors' claims, I'm concerned that the use of ENAS may be superfluous, and that equally good results could be obtained by skipping ENAS (Line 11 of Algorithm 1) and always selecting the largest possible architecture in the search space. This is because (a) larger models, properly regularized, are often more accurate than smaller ones, and (b) based on the description of ReLU pruning in Section 3.1, I don't think that increasing the number skip connections in the network should lead to an increase in ReLUs. Nor should increasing the convolutional kernel sizes. The best way to address this comment would be for the authors to run a control experiment where they always select the largest possible architecture in the search space instead of running ENAS. If accuracies are on par with the paper's current results, getting rid of ENAS would allow them to drastically simplify their method (but would likely require a major revision). 2) Additional baselines would further strengthen the paper. For example: the authors currently compare the relative accuracies of network architectures obtained with ReLU pruning vs. ReLU shuffling (Section 4.2 and Table 1). To justify these optimizations, it would be helpful to include ReLU count/accuracy numbers for a baseline which includes neither shuffling nor pruning.

Correctness: The authors currently don't currently provide the experiments they'd need to empirically justify the added complexity of using the ENAS architecture search algorithm, which is a major part of their paper. Aside from missing baselines and ablations, the submission's empirical results seem reasonable. I didn't see anything obviously wrong with the cryptographic methods, but I'm also less familiar with them, so I might've missed something.

Clarity: I was confused by Line 9 of Algorithm 1. In the definition of C*_i, how is \beta defined?

Relation to Prior Work: Yes: the submission claims to propose new neural network architectures which are significantly faster than previously proposed architectures when using MiniONN inference.

Reproducibility: Yes

Additional Feedback: I think the high-level approach should be reproducible using the information from the paper, but I'm less sure about the implementation details. To be fully reproducible, more information would need to be provided about the search space that was used in the experiments. (Is this identical to the one in the original ENAS paper or were any modifications made to their search space?) Releasing searched architectures found by ENAS and the code to run experiments would also improve reproducibility..


Review 4

Summary and Contributions: This paper proposes a neural architecture search algorithm, tailored to the specific needs of private inference. The authors rightly point out that due to the constraints of the underlying cryptographic primitives the latency of private inference is largely dominated by the computation of the non-linear activation functions. They present a NAS algorithm that optimizes the Network for accuracy under the constraint of a fixed ReLU budget. The ReLU balancing, shuffling and pruning techniques improve the Pareto frontier of the accuracy/runtime trade-off compared to previous work.

Strengths: Translating algorithms 1-1 into the encrypted domain will not necessarily lead to good results. This work understood that secure computation has different requirements and thus a re-think of how to do inference is required. As the ReLU computation is the major factor for private inference latency, they presented a principled way of finding CNNs with skip connections with reduced ReLU count but comparable accuracy. This is refreshingly novel, as most works on private inference solely focus on latency but not accuracy. It is good to see that the authors tried to achieve SOTA accuracy scores and also clearly showed the trade-off between faster runtime and better accuracy.

Weaknesses: Given that this ReLU optimization "only" leads to a 2x speed up, it would have been nice to also incorporate the ideas of [17]. I don't agree that this work is totally orthogonal, as the accuracy penalty from using a polynomial activation function could potentially mitigated by additional layers, as the computation of a polynomial is more than an order of magnitude faster than computing the ReLU. The experiments are done on Cifar10/100 which are very similar datasets. It would have been nice to see the effects of this optimization on other application domains. I don't fully understand why the authors chose MiniONN for the experiments, as both GAZELLE and DELPHI outperform MiniONN significantly. Seems like an odd choice in a paper where you want to improve latency.

Correctness: To the best of my knowledge, yes.

Clarity: In general, the paper is clearly written. However, it could do with another proof read. E.g.: line 84: MiniONN [is] uses line 120: where C and S keep[s] line 196: after [at?] layers line 203: because [because] line 258: on a relatively slow P100 GPUs. [Plural or singular, you decide] Also, the citation in line 202 should be 17 not 14.

Relation to Prior Work: yes.

Reproducibility: Yes

Additional Feedback: In the impact section, you could also discuss the dark side. Moving the computation into the encrypted domain makes it a lot harder to detect foul play, like inference attacks.

[Author Response · NeurIPS 2020]

We thank the reviewers for their valuable comments and address the main concerns raised in review order.

**R2** **What is the break-down of the runtime, mostly communication or computation? More thorough discussion of runtime of ReLUs:** The runtime of CryptoNAS is dominated by the cost of securely evaluating ReLUs, for which we used the ABY library [3]. Unfortunately, ABY does not break down total runtime by communication and computation and hence we only reported the end-to-end runtime. We will instrument ABY to measure the two separately and report breakdowns in the appendix. It is worth noting that reducing ReLU counts translates directly in reductions in *both* communication and computation costs, since the bottleneck for both are ReLU evaluations. A discussion of the relative costs will be added to the paper.

**R3** **Better reference for garbled circuits is Bellare et al. at CCS '12:** We thank our reviewer; citation will be fixed.

**R3, R4** **In Algorithm 1 how is $\beta$ defined?** This was an error on our part. In line 198 $\beta$ was missed, it should be $\alpha^2 = \beta = 4$, where $\beta$ is the channel scaling across layers (conventionally set to 2).

**R4** **The use of ENAS may be superfluous. Larger models, properly regularized, are often more accurate:**

We thank the reviewer for this suggestion. We ran the suggested control experiments and the results are shown in Table 1. We observe that the largest network ($5 \times 5$ filters and all skips) for CNet3 ReLU balanced had an accuracy loss of 0.24% (CiFAR-10) and 0.48% (CiFAR-100) compared to the models found by ENAS. That is, the models returned by ENAS are more accurate than the largest all-skip models. This also

Table 1: Comparing models discovered by ENAS to largest models in search space (all-skip, $5 \times 5$ filters).

| Model | ENAS Arch | | Largest Arch | |
|---|---|---|---|---|
| (Base-Dataset) | Params | Acc | Params | Acc |
| CNet3-C10 | 166M | 95.55% | 311M | 95.31% |
| CNet3-C100 | 149M | 79.59% | 311M | 79.11% |

reaffirms the results reported in [26], where models with all possible skip connections showed an accuracy drop.

We note that the models (including largest all-skip models reported in Table 1) are already regularized using dropout and L2 regularization. While it is possible that there is an even better regularizer for the largest model, this would be a significant research problem in itself. In this context, ENAS search can itself be viewed as akin to a regularizer on the largest model since it drops a subset of its skip connections to increase accuracy.

**R4** **Baselines accuracy and ReLU counts of models without shuffling and pruning (Section 4.2 and Table 1):** The ReLU counts for the baseline models (385K, 1.92M, and 3.89M) were reported in the text on Page 6, line 241 but we will add these to Table 1 to highlight them. The accuracy of the baseline is the same as the model with shuffling.

**R4** **The authors currently don't provide the experiments to empirically justify the added complexity of using ENAS:** We thank the reviewer for this point, and will add the control experiments to our paper (as noted in Table 1).

**R4** **Is the search space identical to the one in the original ENAS paper? Releasing searched architectures and code would also improve reproducibility:** Yes, the search space is the same as ENAS' macro-search space. We clarify this in the paper and publicly release our architectures and code.

**R5** **This work is not totally orthogonal to [15], it would have been nice to also incorporate those ideas:** We agree with the reviewer that CryptoNAS and Delphi's [15] optimizations are not entirely orthogonal. Our intent was to note that Delphi can be applied on top of the models found by CryptoNAS for further benefit. But, as the reviewer notes, the two can also be combined in more sophisticated ways which would be an interesting direction for future research. We will update the paper with this note.

**R5** **The experiments are done on Cifar10/100 which are very similar datasets. It would have been nice to see the effects of this optimization on other application domains:** Most prior work on private inference experiments on easier datasets such as MNIST and Cifar10. We included the more challenging Cifar100 dataset in our experiments (Delphi is the only other work to do this), but we concur that experiments on a more diverse range of application domains will be very valuable. We will seek to do this in future work and add a note to this effect in the paper.

**R5** **Why choose MiniONN for the experiments, as both Gazelle and Delphi outperform MiniONN significantly:** The cost of privately computing ReLUs is the shared bottleneck in all three frameworks, and therefore CryptoNAS' benefits will transfer equally to Delphi and Gazelle. We could not use the Delphi protocol since the paper was published only recently and concurrently with our research. Therefore, updating CryptoNAS' implementation from scratch with the Delphi crypto protocol would not have been possible in the available time (although, as shown in Fig. 3, we were able to estimate the runtime of Delphi's architectural optimization using the MiniONN crypto protocol). However, since Delphi crypto protocol improves upon MiniONN's, the runtimes of the CryptoNAS models using Delphi's protocol should be even smaller.

**R5** **In the impact section, you could also discuss the dark side:** We thank the reviewer for this point and will discuss it in the impact section.

[Meta-Review · NeurIPS 2020]

A nice paper on an important topic. The paper had very strong specialist advocates and its content has been recognised as very substantial with very convincing results.